# Quality of Single-Cone Obturation Using Different Sizes of Matching Gutta-Percha Points of Two Reciprocating Single-File Systems in Curved and Straight Root Canals

**DOI:** 10.3390/medicina61030465

**Published:** 2025-03-07

**Authors:** Shakiba Arvaneh, René Schwesig, Shahpar Haghighat, Christian Ralf Gernhardt

**Affiliations:** 1University Outpatient Clinic for Conservative Dentistry and Periodontology, Medical Faculty, Martin-Luther-University Halle-Wittenberg, Magdeburger Str. 16, 06112 Halle, Germany; shakibaarvanehh@gmail.com; 2Department of Orthopaedic and Trauma Surgery, Martin-Luther-University Halle-Wittenberg, Ernst-Grube-Str. 40, 06120 Halle, Germany; rene.schwesig@uk-halle.de; 3ACECR Academic Center for Education, Culture and Research, Breast Cancer Research Center, Motamed Cancer Institute, No 146, South Gandi, Haghani St., Vanak Sq., Teheran P.O. Box 1517964311, Iran; haghighat@acecr.ac.ir

**Keywords:** curved canals, endodontic treatment, matching gutta-percha points, root canal obturation, single-cone technique, single-file systems, reciprocating file

## Abstract

*Background and Objectives:* Endodontic success depends on eliminating infection and creating a durable seal to prevent recontamination. The goal of this study was to assess the impact of different ISO sizes on the obturation quality using two reciprocating single-file systems, WaveOne^®^ Gold and Procodile^®^, in two different canal morphologies. *Material and Methods:* Overall, 140 root canals from human permanent teeth were randomly assigned to 14 groups based on selected ISO sizes and straight and curved canal curvatures, and the two file systems, WaveOne^®^ Gold files in ISO sizes 20, 25, and 45, and Procodile^®^ files in ISO sizes 20, 25, 40, and 45, were employed for canal preparation. These 140 canals were obturated using corresponding gutta-percha points and AH-Plus sealer and the quality of the obturation was assessed after sectioning the roots (apical, middle, coronal third) by evaluating the resulting 420 sections under a digital fluorescence microscope with regard to the proportion of gutta-percha, sealer, and unfilled areas. The results were analyzed using nonparametric tests. *Results:* For both systems, there was a significant difference in the percentage of gutta-percha-filled areas (PGFA, *p* < 0.001) and sealer-filled areas (PSFA, *p* < 0.001 among the different ISO sizes). However, no significant difference was observed in the percentage of unfilled areas (PUA, *p* = 0.354). ISO 40 demonstrated the best results, with the highest percentage of gutta-percha-filled areas (87%) and the lowest percentages of sealer-filled areas (13%) and voids (0.5%). In contrast, the lowest percentages of gutta-percha filled areas were observed in root canal fillings with ISO 20 (81%) and ISO 25 (81%). Regarding both reciprocating file system sizes, ISO 45 in WaveOne^®^ Gold and ISO 40 in Procodile^®^ demonstrated significantly improved (*p* < 0.05) filling quality, with PGFA of 85% and 87%, respectively. The differences between both systems were not significant. *Conclusions:* The results presented suggest that larger sizes provide better filling results, especially in the apical region. These results underline the importance of selecting appropriate preparation sizes adjusted to the initial anatomical specifications to optimize root canal obturation and ensure a high quality and durable seal.

## 1. Introduction

The success of endodontic treatment largely hinges on effectively eliminating infection from the root canal at the outset and preventing recontamination throughout the treatment process [1]. Achieving a durable and hermetic seal in root canal therapy is fundamental to the long-term success of endodontic treatment. This seal is typically established after disinfection and root canal preparation through the obturation of the root canal system, which prevents bacterial recontamination and facilitates periapical healing [2,3,4,5].

Techniques such as cold lateral condensation, thermoplastic obturation and, more recently, the single-cone technique have been employed to achieve this goal [6]. In this context, previous studies have shown that, compared to cold lateral condensation, thermoplastic techniques demonstrate improved adaptability along the canal walls and reduced voids, particularly in irregularly shaped root canal systems [7,8,9]. The single-cone technique involves using a gutta-percha cone, matched to the final shape of the prepared canal, along with a sealer to fill the root canal space [10]. This obturation technique has gained increasing attention due to its simplicity and efficiency in combination with modern endodontic single-file systems. Studies have shown that porosity is comparable or lower with single-cone obturation than with other techniques and that this approach achieves similar success in terms of filling quality and the prevention of apical leakage compared to more complex methods [3,4,11,12,13,14].

Sealers are essential to form an impervious barrier between the core material and the root canal walls and penetrate the dentinal tubules [4,15]. Calcium silicate-based sealers and epoxy resin-based sealers are among the most extensively studied for their effectiveness in conjunction with various obturation techniques [6]. Calcium silicate-based sealers, like Endosequence BC and MTA-based options, have biocompatibility and antibacterial properties and create a robust bond by setting in the presence of moisture, reducing microleakage in complex canals. Resin-based sealers like AH Plus are recognized for their durability and adaptability, especially in lateral condensation techniques [16,17]. Although they can shrink slightly during curing, AH Plus sealers remain popular for their reliable seal when used together with numerous different obturation techniques [18,19]. Although silicate-based sealers are now recommended in many cases calcium in combination with the single-cone obturation technique, AH Plus is also used for the single-cone obturation technique [17,20,21,22].

The morphology of root canals can show a high degree of variability [23]. Therefore, anatomy influences the quality of the obturation, as variations in the canal shape and curvature require customized approaches to ensure an effective seal. This is particularly relevant for single-cone obturation, where the fit of the gutta-percha cone to the internal structure of the canal is crucial to prevent leakage. There is little research investigating how different ISO preparation sizes affect filling quality in root canals with different morphologies, such as curved or straight canals [24,25,26].

This study aimed to investigate the effect of selected ISO sizes on obturation quality when using a single-file system in combination with its corresponding single-cone gutta-percha, employing the single-cone technique. The analysis focuses on the proportions of gutta-percha-filled areas (PGFA), sealer-filled areas (PSFA) and unfilled areas (PUA) at three different levels of curved and straight root canals. This was evaluated using two reciprocating single-file systems, the Procodile^®^ system (Komet Dental Gebr. Brasseler GmbH & Co. KG, Lemgo, Germany) and the WaveOne^®^ system (Dentsply Sirona Deutschland GmbH, Bensheim, Germany), for preparation and utilizing the delivered matching gutta-percha points for single-cone obturation. The hypothesis is that varying the preparation size using the two file systems will significantly affect filling quality, leading to differences in the proportions of gutta-percha, sealer, and unfilled areas across various canal sections and configurations. The null hypothesis suggests that preparation size does not affect filling quality, resulting in consistent percentages of gutta-percha, sealer, and unfilled areas across all experimental groups.

## 2. Materials and Methods

### 2.1. General Study Design

This ex vivo study utilized 140 root canals from freshly extracted human permanent teeth, including both single-rooted and multi-rooted specimens. The teeth were removed in our department for periodontal and orthodontic reasons. Before extraction, patients of all ages (18–65 years old) were informed about the use of their extracted teeth and written consent was obtained. In the case of multirooted teeth, the roots were separated. The selected teeth had not undergone prior endodontic treatment, exhibited completed root development, and had intact root structures. During the experimental procedures, all teeth and prepared samples were stored in sterile 0.9% saline solution (Fresenius Kabi Deutschland GmbH, Bad Homburg, Germany) at room temperature to maintain hydration. Ethical approval for the study protocol and the use of extracted teeth was granted by the Ethics Committee of Martin Luther University Halle-Wittenberg, Halle, Germany (protocol number: 2024-023). Before tooth extraction, all patients were provided with detailed verbal and written information regarding the study and gave their informed consent through signed documentation.

### 2.2. Sample Selection

Teeth with conditions such as root caries, fractures, previous endodontic treatments, or apical resections were excluded from the analysis. For single-rooted teeth, the crowns were removed (decoronated), while multi-rooted teeth were sectioned into individual roots using a diamond bur to facilitate an initial visual examination of the root canal morphology. Finally, all roots were decoronated to standardize the root length to 14 mm. Radiographic imaging was used to evaluate the root canal anatomy, the initial apical file size, working length, and root curvature with sterile K-Files and Hedstrom Files (#8–#30, VDW GmbH, Munich, Germany). All root canals with an initial apical file size exceeding ISO 30 were excluded to reduce anatomical variability and maintain consistency in root canal configurations. Root curvature was classified using Schneider’s method [27], categorizing canals as straight (0°–5°) or curved (10°–20°). Patients displaying canals with curvatures greater than 20° and patients with two canals, oval or irregular shapes, or obliterated canals were also excluded to ensure standardization. To ensure that standardization can be achieved, root canals with an initial apical size of ISO 10 were assigned to the groups prepared with instruments of size ISO 20 and root canals with an initial size up to #15 were assigned to the groups prepared with instruments of ISO 25. The canals with an initial size up to #30 were assigned to the groups prepared with instruments of size 40 and 45. All other root canals were excluded.

### 2.3. Sample Population

After sectioning multi-rooted teeth, a total of 140 root canals were randomly divided into 14 groups based on selected ISO sizes, canal curvature, and the file system. We employed WaveOne^®^ Gold files in ISO sizes 20, 25, and 45 and Procodile^®^ files in ISO sizes 20, 25, 40 and 45. Root canals were selected according to their initial apical file (IAF) compatibility with the respective ISO sizes: canals with an IAF up to #10 were prepared with ISO 20 files, those up to #15 were prepared with ISO 25 files, those up to #25 were prepared with ISO 40 files, and those up to #30 were prepared with ISO 45 files. This ensured optimal adaptation between the file system and the canal’s morphology [28].

Following the manufacturers’ guidelines, each single-file system was used to prepare 20 root canals, with an equal split of 10 curved and 10 straight canals for each ISO size. The prepared canals were then obturated using their corresponding gutta-percha points in combination with AH Plus sealer (Dentsply Sirona Deutschland GmbH, Bensheim, Germany). The quality of obturation was evaluated in three segments of the roots—apical, middle, and coronal—resulting in 420 sections needing to be analyzed (Figure 1).

### 2.4. Root Canal Preparation and Obturation

The preparation process for all 14 groups followed a reciprocating technique using a calibrated endodontic motor (VDW GmbH, Munich, Germany), as per the manufacturers’ protocols. The instruments employed included Procodile^®^ and WaveOne^®^ Gold; both are single-file nickel-titanium systems with different tapers. Both systems were used in a reciprocating motion. In both systems, files with apical sizes of 20, 25, and 40 were available. Additionally, a size 45 file from the Procodile^®^ system was evaluated. In the Procodile^®^ system, ISO 20 and 25 files have a constant taper of 6%, the ISO 40 file has a 5% taper, and the ISO 45 file has a 4% taper. Regarding the WaveOne^®^ Gold files, the ISO 20 and 25 files have a variable taper of 7%, and the ISO 45 file has one of 5%. A new file was used for each canal to ensure standardized conditions across all specimens.

During preparation, sterile 0.9% sodium chloride solution (Fresenius Kabi Deutschland GmbH, Bad Homburg, Germany), 3% sodium hypochlorite solution (NaOCl) (Aug. Hedinger GmbH & Co. KG, Stuttgart, Germany) and 20% ethylenediaminetetraacetic acid (EDTA) (Speiko—Dr. Speier GmbH, Bielefeld, Germany) were used alternately. For clinical relevance, 15 mL of NaOCl and 5 mL of EDTA were used per canal during preparation. The final sonic activation of the irrigation solution was performed using EDDY™ (VDW GmbH, Munich, Germany) [29]. Canals were subsequently dried with paper points (Coltène/Whaledent GmbH, Altstätten, Switzerland). The canals were then obturated using the corresponding prefabricated, size- and taper-compatible gutta-percha points and AH Plus sealer (Dentsply Sirona Deutschland GmbH, Bensheim, Germany) through a single-cone technique. The sealer was applied evenly using a Lentulo spiral (VDW Root Filler, VDW GmbH, Munich, Germany) [30]. Afterward, canal orifices were sealed with Tetric Evo Flow^®^ composite (Ivoclar Vivadent GmbH, Ellwangen, Germany) and the samples were stored in a sodium chloride solution for at least 24 h to allow the complete setting of the root canal fillings before further evaluation (Figure 2).

### 2.5. Evaluation and Statistical Analysis

After the samples were cured, they were embedded in Technovit^®^ resin (Kulzer GmbH, Hanau, Germany) for structural stabilization. Thin slices, each 1 mm thick, were sectioned from the root canals at 3 mm intervals from the apex (i.e., at 3 mm, 6 mm, and 9 mm intervals). This procedure was carried out using a high-precision diamond band saw (EXAKT Advanced Technologies GmbH, Norderstedt, Germany). The prepared sections were then examined using a fluorescence microscope (Compact, model series BZ-X, KEYENCE Deutschland GmbH, Neu-Isenburg, Germany) at a magnification of 6.0x (Objective Lens: PlanApo 4.0 plus digital zoom 1.5, KEYENCE Deutschland GmbH, Neu-Isenburg, Germany). This made it possible to measure and evaluate the areas filled with gutta-percha, the areas covered with sealers, and the unfilled regions (Figure 3a,b).

A power calculation with G*Power 3.1.9.7 for Windows (Heinrich-Heine University, Düsseldorf, Germany) was performed in order to calculate the minimum sample size required to detect relevant differences between both main groups (instruments) regarding the root canal morphology and section localization. Based on the main parameter *t*-test for independent groups, we required a mean difference of 5.0 (85 vs. 80, pooled SD: 9.0; d = 0.56), a significance level of 5%, a power of 80%, and a sample size of 52 samples in each group. Therefore, 52 samples were included in each instrument group. In the Procodile group, 20 samples were additionally included due to the four sizes available in the system.

The statistical analysis of the collected data was assessed using SPSS version 28.0 for Windows (IBM, Amonk, NY, USA). The Kolmogorov–Smirnov test was used to assess the data’s normal distribution. The differences between the three groups were analyzed using the Kruskal–Wallis test, while the Mann–Whitney U test was used for pairwise group comparisons. The significance level was defined with *p* < 0.05.

## 3. Results

### 3.1. General Result per ISO Sizes

The Kolmogorov–Smirnov test showed that the PGFA, PSFA, and PUA did not have normal distributions (*p* < 0.001). Therefore, the data were analyzed using nonparametric tests. According to the Kruskal–Wallis test, there was a significant difference in the percentages of gutta-percha-filled areas (PGFA, H_value = 18.17, df = 3, *p* < 0.001) and sealer-filled areas (PSFA, H_value = 16.68, df = 3, *p* < 0.001) among the different ISO sizes. However, no significant difference was observed in the percentage of unfilled areas (PUA, *p* = 0.354). ISO 40 demonstrated the best results, with the highest percentage of gutta-percha-filled areas (87%) and the lowest percentages of sealer-filled areas (13%) and voids (0.5%). The lowest PGFA was observed in root canal fillings using ISO 20 with a percentage of 81%, followed closely by ISO 25 at 81% (Table 1, Figure 4).

### 3.2. Obturation Quality in Different ISO per File-System

There are significant differences in PGFA and PSFA between the applied ISO sizes in both file systems, WaveOne^®^ Gold (H_value = 9.36, df = 2, *p* = 0.009 for PGFA and H_value = 10.93, df = 2, *p* = 0.004 for PSFA) and Procodile^®^ (H_value = 15.28, df = 3, *p* = 0.002 for PGFA and H_value = 13.14, df = 3, *p* = 0.004 for PSFA). ISO 45 in WaveOne^®^ Gold and ISO 40 in Procodile^®^ demonstrated the best filling quality, with PGFA of 85% and 87%, respectively (Table 2, Figure 5).

According to the Mann–Whitney test, the significant differences between the ISO groups can be attributed to notable differences between ISO 20 and 25 (U_value = 1281, Z_Score = −2.72, *p* = 0.006 for PGFA and U_value = 1264, Z_Score = −2.81, *p* = 0.005 for PSFA), as well as between ISO 25 and 45 (U_value = 1312, Z_Score = −2.56, *p* = 0.010 for PGFA and U_value = 1247, Z_Score = −2.90, *p* = 0.004 for PSFA), in the WaveOne^®^ Gold system. Similarly, significant differences were found between ISO 20 and 40 (U_value = 1178.5, Z_Score = −3.26, *p* = 0.001 for PGFA and U_value = 1215.5, Z_Score = −3.07, *p* = 0.002 for PSFA), ISO 25 and 40 (U_value = 1234.5, Z_Score = −2.97, *p* = 0.003 for PGFA and U_value = 1288.5, Z_Score = −2.68, *p* = 0.007 for PSFA) and ISO 40 and 45 (U_value = 1197, Z_Score = −316, *p* = 0.002 for PGFA and U_value = 1243, Z_Score = −2.92, *p* = 0.003 for PSFA) in the Procodile^®^ system (Table 3).

### 3.3. Obturation Quality in Different ISO per Curvacure

Within both curved and straight canals, significant differences in filling quality were observed for the tested ISO sizes. ISO 40 demonstrated the best results, achieving 88% PGFA, 12% PSFA, and 0.60% PUA in curved canals, and 86% PGFA, 13% PSFA, and 0.34% PUA in straight canals. ISO 45 had the second-best filling quality in both curved and straight canals, with 84% and 85% PGFA, respectively. The poorest results were observed with ISO 20 in curved canals, yielding 79% PGFA and 20% PSFA. In straight canals, ISO 25 showed the lowest performance with 82% PGFA and 17% PSFA, followed closely by ISO 20, which achieved 83% PGFA and 16% PSFA (Table 4, Figure 6).

When analyzing the canal configuration within the ISO sizes examined, we observed that the canal configuration had no significant influence on the filling quality. However, straight configurations consistently delivered better results at most ISO sizes, with the difference becoming more noticeable as the ISO size decreased (Table 5, Figure 6).

### 3.4. Obturation Quality in Different ISO per Canal Section

In all sections of the root canals, PGFA and PSFA showed significant differences. In the apical section, ISO 45 delivered the best results with 83% PGFA and 16% PSFA, while ISO 20 performed the poorest, with 70% PGFA and 29% PSFA. In the middle section, ISO 40 provided the highest quality with 89% PGFA and 11% PSFA, while ISO 25 had the lowest performance at 81% PGFA and 18% PSFA. In the coronal section, ISO 40 achieved the best outcomes, with 91% PGFA and 9% PSFA, followed by ISO 20, with 89% PGFA and 11% PSFA. The least favorable results in the coronal section were observed with ISO 45, showing 84% PGFA and 15% PSFA. No significant differences were observed between the investigated ISO sizes regarding voids (PUA) in all sections of the root canals (Table 6).

## 4. Discussion

In the present study, we investigated the influence of different preparation sizes on the obturation quality of curved and straight root canals with two reciprocal single-file systems, WaveOne^®^ Gold and Procodile^®^, in conjunction with the single-cone obturation technique using the matching and delivered gutta-percha points by evaluating the proportions of gutta-percha-filled areas (PGFA), sealer-filled areas (PSFA) and unfilled areas (PUA) in three root canal sections. There were significant differences in PGFA and PSFA between the different preparation sizes used, with both reciprocating single-file systems, in different canal configurations and canal sections. However, the results of the present study showed that the overall percentage of gutta-percha was considerably high. This means that the choice of the correct size for canal preparation with single-file systems and their subsequent obturation, with their matching gutta-percha tips adapted to the initial canal anatomy, is crucial for the resulting root canal’s filling quality. Regarding the results presented, the null hypothesis that the prepared size has no influence on the filling quality, meaning that the percentage of gutta-percha, sealer, and unfilled areas is the same in all groups, therefore had to be rejected.

The treatment steps for sample preparation, the application of all materials, and the following evaluation of the slices under the microscope were performed by a single researcher to avoid any interference by other persons. This researcher was trained using the file systems, obturation materials, and all steps regarding the study protocol in advance. This allows for a high degree of standardization. The endodontic treatment of the samples was performed according to a clinically recommended protocol that included all necessary clinical steps [5]. Regarding the chemical disinfection protocol during root canal preparation, sodium hypochlorite and ethylenediaminetetraacetic acid (EDTA) were alternately used as disinfectants without mixing them [31,32]. The additional performed passive sonic activation during irrigation follows a clinical recommended protocol and reduced the risk influencing the obturation quality [21]. Between each irrigation step, the irrigation solutions were removed using paper points. Using the solutions this way, the known interaction between both irrigation solutions could be avoided using this protocol [33]. Thus, the possible risk of sealing dentinal walls with a developing precipitate is unlikely to have any importance [33]. In the present study, AH Plus was used as a sealer. Sealers are essential, forming an impervious barrier between the core material and the root canal walls and penetrating the dentinal tubules [4,15]. Regarding our irrigation protocol, it should have no influence on the sealing ability of AH Plus [16,34]. Calcium silicate-based sealers and epoxy resin-based sealers are among the most extensively studied for their effectiveness in conjunction with single-cone obturation techniques [6]. In this context, the use of bioceramic sealers has been recommended in some studies to reduce voids and enhance the density of the apical foramen [35,36,37,38,39,40]. Calcium silicate-based sealers, like Endosequence BC and MTA-based options, have biocompatibility and antibacterial properties and create a robust bond by setting in the presence of moisture, reducing microleakage in complex canals. Resin-based sealers like AH Plus are recognized for their durability and adaptability, especially in lateral condensation techniques [16,17]. Although they can shrink slightly during curing, AH Plus sealers remain popular for their reliable seal [18,19,34]. The irrigation protocol using sonic activation allows AH Plus to penetrate the dentinal tubules [21]. The present results using AH Plus are comparable to those in other studies using different sealers [41].

A previous study showed no significant differences between the two reciprocating single-file systems WaveOne^®^ Gold and Procodile^®^ in terms of obturation quality using ISO 25 instruments and gutta-percha points. Both systems delivered comparable results in this respect [11]. This study has shown that while the choice of the company does not cause significant differences, the size leads to significant differences in obturation quality. Higher ISO values, particularly ISO 40 in Procodile^®^ and ISO 45 in WaveOne^®^ Gold, consistently showed better results (PGFA percentages: 87 and 85%). In contrast, smaller sizes, particularly ISO size 20, resulted in poorer outcomes, especially in the apical region and curved root canals. This is in accordance with other investigations into the impact of preparation size [42].

In our study, straight configurations saw obturation perform mostly better than in curved canals, especially using smaller preparation sizes. However, this difference was not statistically significant. Nevertheless, the difference in filling quality between the different ISO sizes was significant in both curved and straight canals. This indicates that obturation quality is influenced more by the ISO file size selected than by the channel’s configuration itself.

The difference in filling quality between the smallest and largest ISO sizes decreases from the apical to the coronal section. While a notable difference in the proportion of gutta-percha-filled areas (PGFA) is observed apically between ISO 20 (70%) and ISO 45 (83%), this difference becomes negligible in coronal parts, with PGFA values of 91% for ISO 40 and 89% for ISO 20. Therefore, ensuring adequate irrigation and disinfection of the apical section is critical for eliminating microorganisms [43,44,45]. Previous studies have indicated that the effective penetration of irrigants into the apical third of the root canal requires at least an ISO 30 file size [46].

Previous studies have demonstrated that microleakage, caused by porosity and voids within root canals after obturation, affects the success of endodontic treatment and periapical healing [45,47,48]. In this study, no influence of preparation size on the percentage of unfilled areas (PUA) was observed. Across all sizes, the PUA remained below 1%, with an average of 0.9%. This finding suggests that the combination of a sealer with exactly matching single-cone gutta-percha may ensure dense obturation, regardless of the preparation size used.

Previous studies have shown that the choice of sealer and sealer placement can influence filling quality [49]. The results of our study indicate that the apical region of the canal has the highest percentage of sealer-filled areas (PSFA), with this proportion increasing as the preparation size decreases. Some studies have shown that the use of instruments such as K-files or Lentulo spirals for the placement of the sealer in the single-cone technique is beneficial for a reduction in voids [49]. This method ensures better distribution of the sealer and minimizes cavities, which contributes to more homogeneous and denser root canal filling [30,50].

This study contributes to the understanding of single-file root canal preparation systems combined with single-cone obturation techniques. The superior performance of ISO 40 and ISO 45 underscores the necessity of selecting appropriate preparation sizes for optimal obturation quality and being informed by the root canal’s anatomy. While canal morphology—root canals with an angulation up to 20° were included in this study—had no significant effect, selecting the correct size played a critical role in ensuring successful outcomes. The clinical value of our results might be limited because root canals with a higher curvature, often difficult to prepare and obturate, were excluded. Furthermore, using higher preparation sizes, it is important to consider that root canal preparation performed with single-file systems, including higher diameters or high-tapered instruments, might initiate dentinal cracks and increase root fracture risk because of thinned dentinal walls [51,52]. This point should be considered in clinical practice. However, the single-cone obturation technique performed without any pressure might lead to a lower risk of root fractures [53].

Despite the present findings, it is important to acknowledge the limitations of this ex vivo study. While efforts were made to standardize procedures, clinical variables such as practitioner skill, tooth-specific anatomical variations, and long-term treatment outcomes were not accounted for. One of the biggest limitations of this study is the evaluation of specific cross-sections using a 2D assessment instead of 3D analysis performed with a micro-CT [54]. Three-dimensional evaluation methods have become popular in recent years [17,22]. We used 2D assessment because it allows comparability with former work and other studies regarding the quality of root canal obturation techniques [20]. However, the 2D assessment only allowed us to analyze three sections of the entire root. This might limit the quality of the results presented. Furthermore, the study did not judge the influence of different investigators (interobserver reliability) concerning the percentage of gutta-percha-filled areas (PGFA), sealer-filled areas (PSFA), and unfilled areas (PUA) as there was only one single observer performing the evaluation of the 420 sections. However, this point might influence the results. Additionally, differences in the taper and diameter of root canals, as well as anatomical variations among the samples, might influence the results. The standardization protocol used in the present investigation is an important factor in this kind of laboratory studies and leads to the high quality of the investigated root canal filling, with a remarkable portion of gutta-percha filled areas. The teeth used in the study were obtained from various sources without consistently considering patient age, potentially introducing variability. Furthermore, samples with complex canal morphologies, such as oval or C-shaped canals, were excluded, limiting the generalizability of the findings. The clinical situation in most cases did not follow the applied standardization protocol. Therefore, this factor should be considered when transferring the results into clinical dentistry. Finally, discrepancies in dimensions between file instruments and their matching gutta-percha cones may have affected the results. Regarding the results, these discrepancies are very small. This is in good accordance with previously published papers regarding the correlation between prepared root canals and corresponding gutta-percha points, especially in the apical third [55]. The strength of the present investigation is the complex research method, allowing a high level of standardization following clinical and recently published protocols. The high number of specimen slices evaluated is one of the major factors of this study. The findings emphasize the critical importance of size selection in response to the anatomical situation, not only with regard to the chemo-mechanical preparation protocol but also with regard to high-quality root canal fillings in clinical practice, particularly in the apical region, where effective sealing is essential for long-term treatment success [55]. Compared to other obturation techniques, single-cone obturation using exactly matching gutta-percha points might be an alternative. Results focusing on this point underline these findings [55]. The quality of single-cone obturation using these two systems and the corresponding gutta-percha points showed that the quality and composition of the root canal filling is sufficient. Future studies should explore the impact of newer and modern bioceramic sealers, as they have shown potential to enhance obturation quality and antibacterial properties. In combination with silicate-based sealer, single-cone obturation showed promising clinical results [56]. In addition, zinc-oxide-eugenol based sealers showed good clinical results when combined with single-cone obturation [57]. Regarding the epoxy resin-based sealer used, AH Plus, similar results are reported [41]. Therefore, the sealer used could be an alternative to calcium-silicate based sealers. However, the impact of this on long-term clinical success remains still unclear [58]. This important point should be addressed in clinical trials.

## 5. Conclusions

The limitations of an in vitro study aside, this study demonstrates that the prepared root canal size significantly influences obturation quality using a single-cone obturation technique, with ISO 40 and 45 producing the best results across both systems tested in various sections and configurations of root canals. The findings emphasize the critical importance of size selection in clinical practice to achieve high-quality root canal fillings, particularly in the apical region, where effective sealing is essential for long-term treatment success. The present results underline the importance of selecting appropriate preparation sizes, adjusted to the initial anatomical specifications, to optimize root canal obturation and ensure a high-quality and durable seal.

## Figures and Tables

**Figure 1 medicina-61-00465-f001:**
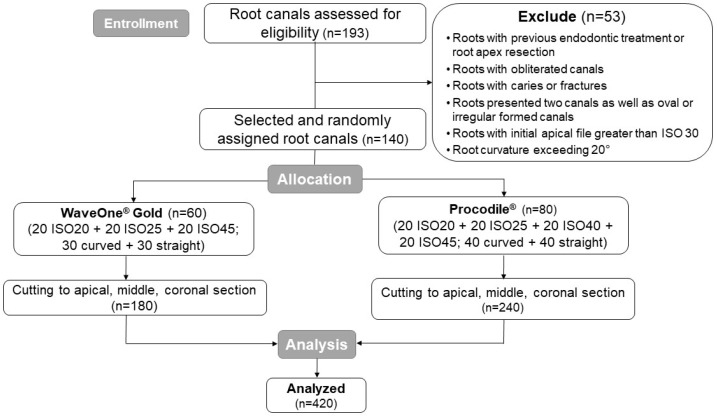
Sample selection and distribution within the different experimental groups.

**Figure 2 medicina-61-00465-f002:**
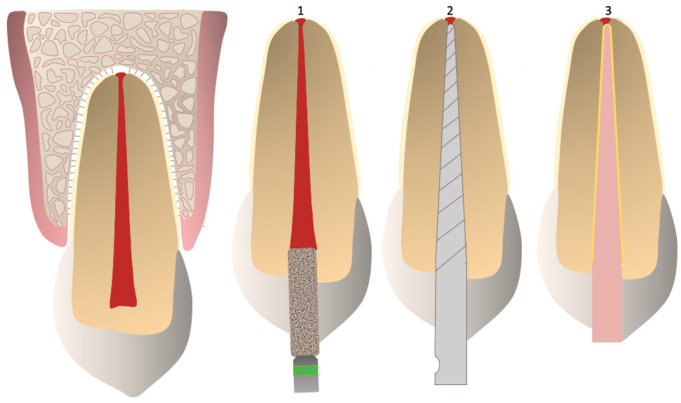
Steps in root canal treatment: 1. trepanation; 2. shaping and cleaning; 3. obturation using a single cone Gutta-percha and sealer.

**Figure 3 medicina-61-00465-f003:**
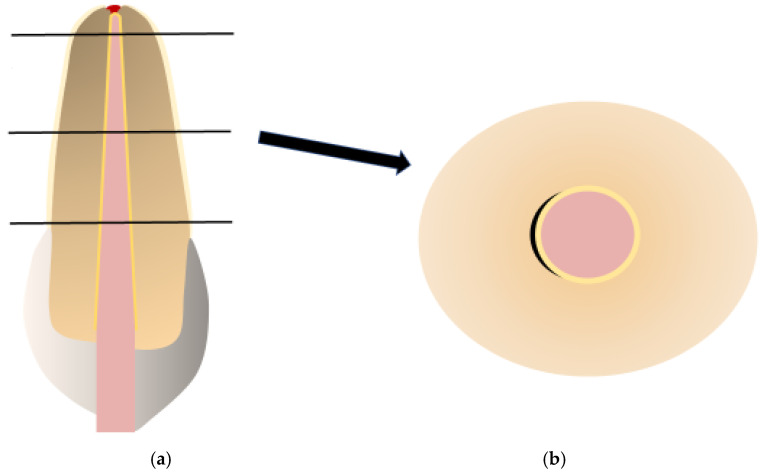
(**a**) The root-filled tooth was sectioned at three levels (apical, middle, and coronal). (**b**) A schematic representation of sections under a microscope (magnification 6×): red—gutta-percha; yellow—sealer; black—voids.

**Figure 4 medicina-61-00465-f004:**
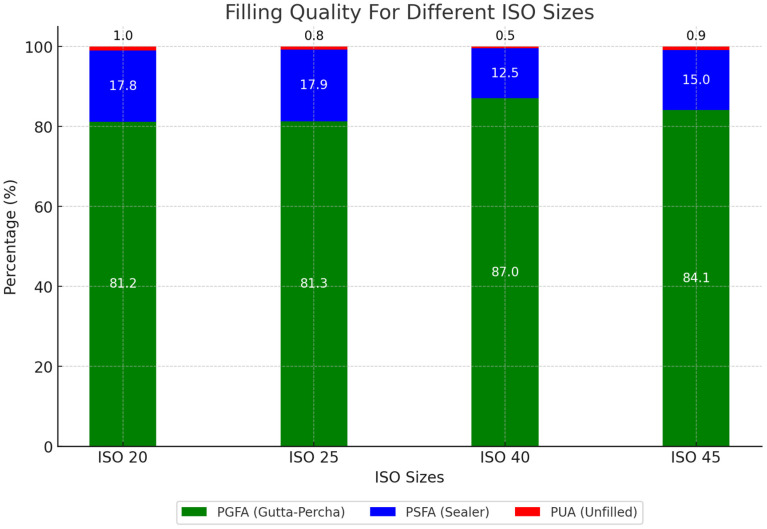
Percentage of gutta-percha, sealer, and unfilled areas regarding different ISO sizes.

**Figure 5 medicina-61-00465-f005:**
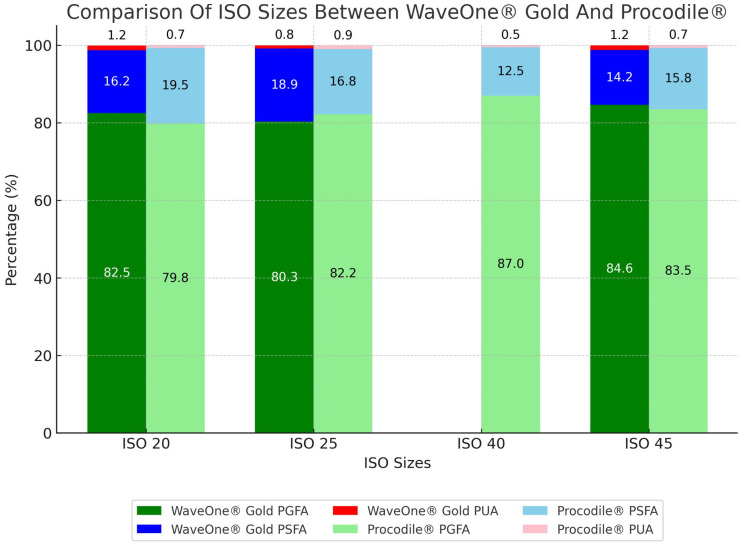
Percentage of gutta-percha, sealer, and unfilled area per ISO size depending on two systems.

**Figure 6 medicina-61-00465-f006:**
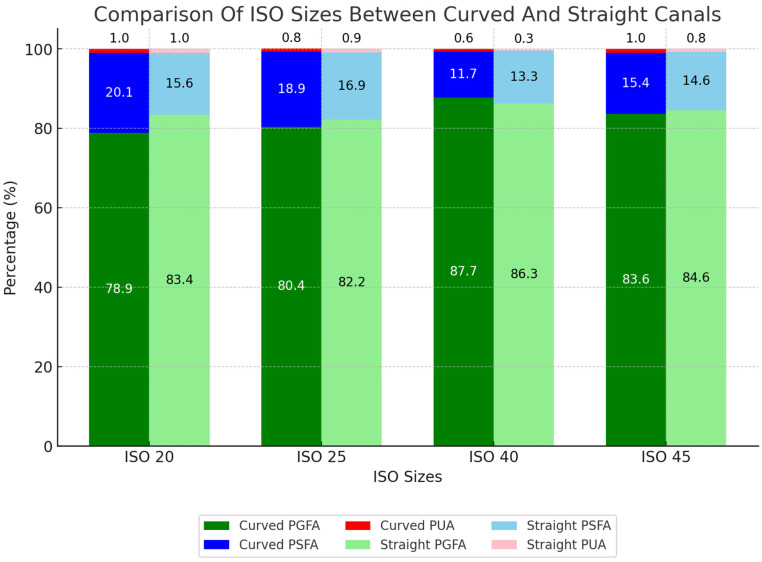
Percentage of areas filled with gutta-percha, sealer, and unfilled area for ISO size depending on curved and straight canals.

**Table 1 medicina-61-00465-t001:** The evaluation of the filling quality between the ISO sizes: SD = standard deviation; 95% CI = 95% confidence interval. mean [%]; PGFA = percentage of gutta-percha-filled areas; PSFA = percentage of sealer-filled areas; PUA = percentage of unfilled areas; only effect sizes d > 0.5 are reported (* Kruskal–Wallis test).

	ISO	n	Mean	SD	95% CI	*p* *	d
PGFA	20	120	81.2	13.6	78.7–83.6	<0.001	25 vs. 40: 0.74 20 vs. 40: 0.59
	25	120	81.3	9.32	79.6–82.9		
	40	60	87.0	6.16	85.4–88.6		
	45	120	84.1	6.89	82.8–85.3		
	Total	420	82.9	10.0	81.9–83.8		
PSFA	20	120	17.8	12.7	15.5–20.2	0.001	25 vs. 40: 0.73 20 vs. 40: 0.56
	25	120	17.9	8.69	16.3–19.5		
	40	60	12.5	6.11	10.9–14.1		
	45	120	15.0	6.82	13.8–16.2		
	Total	420	16.3	9.53	15.4–17.2		
PUA	20	120	0.99	2.51	0.54–1.44	0.354	-
	25	120	0.85	2.22	0.45–1.25		
	40	60	0.47	1.03	0.20–0.74		
	45	120	0.91	1.87	0.57–1.25		
	Total	420	0.85	2.09	0.65–1.01		

**Table 2 medicina-61-00465-t002:** A comparison of the ISO sizes in each file system (* Kruskal–Wallis test). mean [%]; d > 0.5 are reported.

File-System	Area	ISO	n	Mean	SD	95% CI	*p* *	d
WaveOne^®^ Gold	PGFA	20	60	82.5	13.2	79.1–85.9	0.009	25 vs. 45: 0.52
		25	60	80.3	9.03	77.9–82.6		
		45	60	84.6	7.65	82.7–86.6		
		Total	180	82.5	10.4	80.9–84.0		
	PSFA	20	60	16.2	11.5	13.3–19.2	0.004	25 vs. 45: 0.57
		25	60	18.9	8.36	16.8–21.1		
		45	60	14.3	7.73	12.3–16.2		
		Total	180	16.5	9.47	15.1–17.9		
	PUA	20	60	1.24	3.19	0.41–2.06	0.133	-
		25	60	0.77	2.40	0.15–1.38		
		45	60	1.12	2.10	0.58–1.66		
		Total	180	1.04	2.60	0.66–1.42		
Procodile^®^	PGFA	20	60	79.8	14.0	76.2–83.4	0.002	20 vs. 40: 0.71 25 vs. 40: 0.61 40 vs. 45: 0.57
		25	60	82.2	9.57	79.8–84.7		
		40	60	87.0	6.16	85.4–88.6		
		45	60	83.5	6.05	82.8–85.3		
		Total	240	83.1	9.79	81.9–83.8		
	PSFA	20	60	19.5	14.0	15.5–20.2	0.004	20 vs. 40: 0.70
		25	60	16.8	8.96	16.3–19.5		
		40	60	12.5	6.11	10.9–14.1		
		45	60	15.8	5.73	13.8–16.2		
		Total	240	16.2	9.59	15.4–17.2		
	PUA	20	60	0.74	1.55	0.34–1.34	0.833	-
		25	60	0.94	2.05	0.41–1.46		
		40	60	0.47	1.03	0.20–0.74		
		45	60	0.71	1.61	0.29–1.13		
		Total	240	0.71	1.60	0.51–0.92		

**Table 3 medicina-61-00465-t003:** Partial testing depending on area and ISO size (* Mann–Whitney test).

File-System	Area	ISO	Compared ISO	*p* *
WaveOne^®^ Gold	PGFA	20	20 × 25	0.006
			20 × 45	0.871
		25	25 × 45	0.010
	PSFA	20	20 × 25	0.005
			20 × 45	0.904
		25	25 × 45	0.004
Procodile^®^	PGFA	20	20 × 25	0.497
			20 × 40	0.001
			20 × 45	0.407
		25	25 × 40	0.003
			25 × 45	0.946
		40	40 × 45	0.002
	PSFA	20	20 × 25	0.483
			20 × 40	0.002
			20 × 45	0.413
		25	25 × 40	0.007
			25 × 45	0.992
		40	40 × 45	0.003

**Table 4 medicina-61-00465-t004:** Evaluation of ISO sizes in different configurations (* Kruskal–Wallis test). mean [%]; d > 0.5 are reported.

Configuration	Area	ISO	n	Mean	SD	95% CI	*p* *	D
Curved	PGFA	20	60	78.9	14.5	75.2–82.7	0.002	25 vs. 40: 0.88 20 vs. 40: 0.84 40 vs. 45: 0.60
		25	60	80.4	10.2	77.7–83.0		
		40	30	87.7	6.46	85.2–90.1		
		45	60	83.6	7.14	81.7–85.4		
		Total	210	81.9	10.8	80.4–83.4		
	PSFA	20	60	20.1	13.8	16.5–23.6	0.002	20 vs. 40: 0.84
		25	60	18.9	9.32	16.4–21.3		
		40	30	11.7	6.16	9.44–14.0		
		45	60	15.4	6.80	13.7–17.2		
		Total	210	17.2	10.2	15.8–18.6		
	PUA	20	60	1.00	3.11	0.19–1.80	0.309	-
		25	60	0.77	2.51	0.12–1.42		
		40	30	0.60	1.26	0.13–1.07		
		45	60	1.00	1.84	0.52–1.47		
		Total	210	0.88	2.39	0.55–1.20		
Straight	PGFA	20	60	83.4	12.4	80.1–86.6	0.087	25 vs. 40: 0.58
		25	60	82.2	8.36	80.1–84.3		
		40	30	86.3	5.88	84.1–88.5		
		45	60	84.6	6.65	82.9–88.3		
		Total	210	83.8	9.08	82.6–85.0		
	PSFA	20	60	15.6	11.5	12.6–18.6	0.167	25 vs. 40: 0.51
		25	60	16.9	7.97	11.1–15.6		
		40	30	13.3	6.05	12.8–16.4		
		45	60	14.6	6.88	12.8–16.4		
		Total	210	15.4	8.68	14.2–16.6		
	PUA	20	60	0.98	1.75	0.53–1.44	0.432	20 vs. 40: 0.51
		25	60	0.93	1.91	0.44–1.42		
		40	30	0.34	0.74	0.06–0.61		
		45	60	0.83	1.92	0.34–1.33		
		Total	210	0.83	1.75	0.59–1.07		

**Table 5 medicina-61-00465-t005:** Influence of canal curvature on obturation quality in each ISO (* Mann–Whitney test). mean [%].

ISO	Area	Configuration	n	Mean	SD	95% CI	*p* *	d
20	PGFA	Curved	60	78.9	14.5	75.2–82.7	0.073	0.34
		Straight	60	83.4	12.4	80.2–86.6		
		Total	120	81.2	13.6	78.7–83.6		
	PSFA	Curved	60	20.1	13.8	16.5–23.6	0.050	0.36
		Straight	60	15.6	11.5	12.6–18.6		
		Total	120	17.8	12.9	15.5–20.2		
	PUA	Curved	60	1.00	3.11	0.19–1.80	0.253	0.01
		Straight	60	0.98	1.75	0.53–1.44		
		Total	120	1.00	2.51	0.54–1.44		
25	PGFA	Curved	60	80.4	10.2	77.7–83.0	0.447	0.19
		Straight	60	82.2	8.36	80.0–84.3		
		Total	120	81.3	9.32	79.6–82.9		
	PSFA	Curved	60	18.9	9.32	16.4–21.3	0.311	0.23
		Straight	60	16.9	7.97	14.9–19.0		
		Total	120	17.9	8.69	16.3–19.5		
	PUA	Curved	60	0.77	2.51	0.12–1.42	0.137	0.07
		Straight	60	0.93	1.97	0.44–1.42		
		Total	120	0.85	2.22	0.45–1.25		
40	PGFA	Curved	30	87.7	6.46	85.2–90.1	0.359	0.23
		Straight	30	86.3	5.88	84.1–88.5		
		Total	60	87.0	6.16	85.4–88.6		
	PSFA	Curved	30	11.7	6.16	9.44–14.0	0.383	0.26
		Straight	30	13.3	6.05	11.1–15.6		
		Total	60	12.5	6.11	10.9–14.1		
	PUA	Curved	30	0.60	1.26	0.13–1.07	0.637	0.26
		Straight	30	0.34	0.74	0.06–0.62		
		Total	60	0.47	1.03	0.20–0.74		
45	PGFA	Curved	60	83.6	7.14	81.7–85.4	0.512	0.15
		Straight	60	84.6	6.65	82.8–85.3		
		Total	120	84.1	6.89	82.8–85.3		
	PSFA	Curved	60	15.4	6.79	13.7–17.2	0.700	0.12
		Straight	60	14.6	6.88	12.8–16.4		
		Total	120	15.0	6.81	13.8–16.2		
	PUA	Curved	60	1.00	1.84	0.52–1.47	0.612	0.09
		Straight	60	0.83	1.91	0.34–1.33		
		Total	120	0.91	1.87	0.58–1.25		

**Table 6 medicina-61-00465-t006:** Evaluation of obturation quality in different sections (* Kruskal–Wallis test). mean [%]; d > 0.5 are reported.

Configuration	Area	ISO	n	Mean	SD	95% CI	*p* *	d
Apical	PGFA	20	40	69.7	17.1	64.2–75.1	<0.001	20 vs. 40: 1.07 20 vs. 25: 0.62 25 vs. 45: 0.60 20 vs. 45: 0.51 25 vs. 40: 0.51
		25	40	77.9	9.44	74.8–80.9		
		40	20	81.6	5.10	79.2–84.0		
		45	40	83.1	7.94	80.6–85.7		
		Total	140	77.6	12.6	75.5–79.7		
	PSFA	20	40	28.8	16.0	23.7–33.9	<0.001	20 vs. 45: 1.05 20 vs. 40: 0.86 25 vs. 45: 0.65 25 vs. 40: 0.56
		25	40	21.7	8.90	18.8–24.5		
		40	20	17.9	4.72	15.7–20.1		
		45	40	16.3	7.75	13.8–18.7		
		Total	140	21.6	11.8	19.6–23.6		
	PUA	20	40	1.54	3.84	0.31–2.77	0.327	–
		25	40	0.50	1.79	−0.07–1.07		
		40	20	0.52	1.19	−0.04–1.07		
		45	40	0.60	1.62	0.08–1.12		
		Total	140	0.83	2.48	0.41–1.24		
Middle	PGFA	20	40	85.3	6.35	83.2–87.3	0.001	25 vs. 40: 1.15 40 vs. 45: 0.65 20 vs. 25: 0.61 20 vs. 40: 0.58 25 vs. 45: 0.53
		25	40	80.7	8.80	77.9–83.6		
		40	20	88.5	4.77	86.2–90.7		
		45	40	84.8	6.63	82.6–86.9		
		Total	140	84.3	7.41	83.1–85.5		
	PSFA	20	40	14.0	6.28	12.0–16.0	0.001	25 vs. 40: 1.14 25 vs. 45: 0.59 20 vs. 25: 0.59 20 vs. 40: 0.52
		25	40	18.1	7.55	15.7–20.5		
		40	20	11.2	4.59	9.07–13.4		
		45	40	13.9	6.65	11.4–16.1		
		Total	140	14.8	6.91	13.6–15.9		
	PUA	20	40	0.71	1.28	0.30–1.12	0.380	–
		25	40	1.13	2.75	0.25–2.01		
		40	20	0.29	0.71	−0.43–0.62		
		45	40	1.30	2.42	0.52–2.07		
		Total	140	0.94	2.10	0.58–1.29		
Coronal	PGFA	20	40	88.6	4.91	87.0–90.1	<0.001	40 vs. 45: 1.26 25 vs. 40: 0.89 20 vs. 25: 0.51
		25	40	85.2	8.37	82.5–87.9		
		40	20	90.9	4.45	88.8–93.0		
		45	40	84.3	6.04	82.3–86.2		
		Total	140	86.7	6.72	85.6–87.8		
	PSFA	20	40	10.7	4.44	9.29–12.1	<0.001	40 vs. 45: 1.20 25 vs. 40: 0.85 20 vs. 45: 0.81
		25	40	13.9	7.95	11.3–16.4		
		40	20	8.49	4.81	6.24–10.7		
		45	40	14.9	5.89	13.0–16.7		
		Total	140	12.5	6.44	11.4–13.6		
	PUA	20	40	0.72	1.55	0.23–1.22	0.874	–
		25	40	0.93	2.02	0.28–1.58		
		40	20	0.61	1.15	0.07–1.15		
		45	40	0.84	1.41	0.39–1.29		
		Total	140	0.80	1.60	0.53–1.07		

## Data Availability

The data are available upon request from the corresponding author.

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
