# Peer review of "Quality of Single-Cone Obturation Using Different Sizes of Matching Gutta-Percha Points of Two Reciprocating Single-File Systems in Curved and Straight Root Canals"

_medicina, 2025, doi:10.3390/medicina61030465_

Round 1

Reviewer 1 Report

Comments and Suggestions for Authors

Thank you for submitting your manuscript titled “Obturation Quality of Single-Cone Obturation Using Different Sizes of Matching Gutta-Percha Points of Two Reciprocating Single-File Systems in Curved and Straight Root Canals” for peer review. Your study addresses an essential aspect of endodontic treatment, specifically the impact of different ISO sizes on obturation quality, and contributes valuable insights into optimizing root canal obturation. However, several areas could benefit from further clarification and expansion to enhance the scientific rigor and clinical relevance of your work.

The methodology is well-detailed, but we recommend adding further clarification regarding the sample size determination. It is not clear whether a power analysis was conducted to establish the minimum sample size required to detect significant differences. If this analysis was not performed, we suggest acknowledging this limitation and discussing its potential impact on the generalizability of the findings. I recommend adding a paragraph in the limitations explaining that the sample is very small.

Additionally, we recommend expanding the section on study limitations. While the manuscript acknowledges the in vitro nature of the research, further discussion of its constraints would strengthen its validity. For example, clinical factors such as operator variability, anatomical differences in patient populations, and the long-term stability of the obturation quality should be discussed. Addressing these aspects will provide a more balanced interpretation of the study's findings.

The manuscript presents interesting results regarding the relationship between ISO size and filling quality. However, we recommend further elaboration on the clinical implications of these findings. How can these results influence decision-making in daily endodontic practice? Should clinicians prefer larger ISO sizes for improved obturation quality in certain cases? A dedicated discussion of the clinical relevance would enhance the practical applicability of your study.

Furthermore, we encourage the authors to consider discussing the potential risks associated with using instruments with increased diameter and taper. A recent systematic review has highlighted that larger diameter instruments may elevate the risk of vertical root fractures. Given that vertical fractures are a major cause of endodontic failure, we suggest incorporating a brief discussion on this potential risk and how clinicians might mitigate it when selecting instrumentation sizes. I recommend adding a paragraph explaining this, as discussed in this review Puleio F, Lo Giudice G, Militi A, Bellezza U, Lo Giudice R. Does Low-Taper Root Canal Shaping Decrease the Risk of Root Fracture? A Systematic Review. Dent J (Basel). 2022 Jun 1;10(6):94. doi: 10.3390/dj10060094. PMID: 35735636; PMCID: PMC9222076.

Overall, your study provides meaningful insights into obturation techniques and root canal filling quality. By addressing the suggested revisions, your manuscript will offer a more comprehensive and clinically applicable resource for the endodontic community. 

Author Response

Obturation Quality of Single-Cone Obturation Using Different Sizes of Matching Gutta-Percha Points of Two Reciprocating Single-File Systems in Curved and Straight Root Canals

Point by point response to all comments raised by Reviewer 1

Dear Reviewer, thank you very much for your review and all valuable comments and recommendations to increase the quality of the submitted paper.

Thank you for submitting your manuscript titled “Obturation Quality of Single-Cone Obturation Using Different Sizes of Matching Gutta-Percha Points of Two Reciprocating Single-File Systems in Curved and Straight Root Canals” for peer review. Your study addresses an essential aspect of endodontic treatment, specifically the impact of different ISO sizes on obturation quality, and contributes valuable insights into optimizing root canal obturation. However, several areas could benefit from further clarification and expansion to enhance the scientific rigor and clinical relevance of your work.

Thank you very much for your feedback and the helpful comments within your review report.

  1. The methodology is well-detailed, but we recommend adding further clarification regarding the sample size determination. It is not clear whether a power analysis was conducted to establish the minimum sample size required to detect significant differences. If this analysis was not performed, we suggest acknowledging this limitation and discussing its potential impact on the generalizability of the findings. I recommend adding a paragraph in the limitations explaining that the sample is very small.

Thanks a lot for this valuable hint. We added the sample size calculation as follows:

Line 174-181:    A power calculation with G*Power (Fault et al. 2007) was performed in order to cal-culate the minimum sample size required to detect relevant differences between both main groups (instruments) no regarding the root canal morphology and section high. Based on the main parameter a t-test for independent groups, a mean difference of 5.0 (85 vs. 80, pooled SD: 9.0; d = 0.56), a significance level of 5% and a power of 80%, a sample size of 52 samples in each group are necessary. Therefore, 52 samples were included in each instrument groups. In the Procodile group 20 samples were additionally included due to the 4 sizes available in this system.

  1. Additionally, we recommend expanding the section on study limitations. While the manuscript acknowledges the in vitro nature of the research, further discussion of its constraints would strengthen its validity. For example, clinical factors such as operator variability, anatomical differences in patient populations, and the long-term stability of the obturation quality should be discussed. Addressing these aspects will provide a more balanced interpretation of the study's findings.

Thank you for this recommendation. The entire discussion section was revised, clarified and rewritten. The mentioned clinical factors were addressed and included. References were added.

Please see the discussion section.

  1. The manuscript presents interesting results regarding the relationship between ISO size and filling quality. However, we recommend further elaboration on the clinical implications of these findings. How can these results influence decision-making in daily endodontic practice? Should clinicians prefer larger ISO sizes for improved obturation quality in certain cases? A dedicated discussion of the clinical relevance would enhance the practical applicability of your study.

Dear Reviewer, this is an interesting and important point. Therefore, the discussion section was corrected, detailed information was added.

Please see the discussion section.

  1. Furthermore, we encourage the authors to consider discussing the potential risks associated with using instruments with increased diameter and taper. A recent systematic review has highlighted that larger diameter instruments may elevate the risk of vertical root fractures. Given that vertical fractures are a major cause of endodontic failure, we suggest incorporating a brief discussion on this potential risk and how clinicians might mitigate it when selecting instrumentation sizes. I recommend adding a paragraph explaining this, as discussed in this review Puleio F, Lo Giudice G, Militi A, Bellezza U, Lo Giudice R. Does Low-Taper Root Canal Shaping Decrease the Risk of Root Fracture? A Systematic Review. Dent J (Basel). 2022 Jun 1;10(6):94. doi: 10.3390/dj10060094. PMID: 35735636; PMCID: PMC9222076.

Thank you very much for this important comment. We added this point to the discussion section and included the recommended publication. Literature and information dealing with the influence of diameter and taper on the risk of root fractures were added.

Please see the discussion section:

Overall, your study provides meaningful insights into obturation techniques and root canal filling quality. By addressing the suggested revisions, your manuscript will offer a more comprehensive and clinically applicable resource for the endodontic community. 

Thank you very much.

Reviewer 2 Report

Comments and Suggestions for Authors

The title and Abstract sounds good

I suggest adding one more keyword to make five of them, and place them by alphabetic order.

From where were the teeth collected. And why were they extracted?

Please place a legend of the abbreviations of Tables 1, 2, 3, 4, 5, 6

The number and letters of Figures 1, 4 and 5 are very small

The option of using single cone technique with AH plus should be explained since this is not the recomended technique for this sealer.

I believe one of the biggest limitations of this study is the evaluation of specific cross-sections and not a tri-dimentional analysis. The option for this 2D assessment instead of a 3D assessment should be explained.

Additionally, this should be mentioned as a study limitation.

The study strenght should be exposed.

The possible generalization of the results should be debated.

Author Response

Obturation Quality of Single-Cone Obturation Using Different Sizes of Matching Gutta-Percha Points of Two Reciprocating Single-File Systems in Curved and Straight Root Canals

Point by point response to all comments raised by Reviewer 2

Dear reviewer, thank you very much for your review and all valuable comments and recommendations to increase the quality of the submitted paper.

  1. The title and Abstract sounds good

Thank you very much for this feedback.

  1. I suggest adding one more keyword to make five of them, and place them by alphabetic order.

Thank you for this comment. Additional keywords were added and organized in alphabetic order.

Please see the keywords section (line 43-44).

  1. From where were the teeth collected. And why were they extracted?

Thank you for this helpful comment. The missing information were added in the Material and Methods section. The missing details were included.

Please see section 2.1. and 2.2.  

  1. Please place a legend of the abbreviations of Tables 1, 2, 3, 4, 5, 6

This comment is correct. Unfortunately, we missed explanation within the legends. We did it as suggested. Please note, we only added all abbreviations in the legend of table 1, because we used the same abbreviations in all tables. This might increase readability of the manuscript.

Please see the legend of table 1.

  1. The number and letters of Figures 1, 4 and 5 are very small.

This is correct. Thank you. We revised the figures and enlarged the numbers and letters.

Please see the correct and newly included figures 1, 4 and 5.

  1. The option of using single cone technique with AH plus should be explained since this is not the recomended technique for this sealer.

Thank you for this comment. We used AH plus because at this time regarding the development of calcium silicate-based materials it is still the Gold Standard in most countries also together with single-cone obturation technique. We added the missing explanation in the discussion section.

Please see the introduction and discussion section. References were added.

  1. I believe one of the biggest limitations of this study is the evaluation of specific cross-sections and not a tri-dimentional analysis. The option for this 2D assessment instead of a 3D assessment should be explained. Additionally, this should be mentioned as a study limitation.

Thank you for this comment and recommendation. We added this limitation to the discussion section.

Line 333-335:   One of the biggest limitations of this study is the evaluation of specific cross-sections using a 2D assessment instead of a 3D analysis. In the present study a 2D assessment was used….

  1. The study strenght should be exposed.

Thank you for this recommendation. These points were included in the discussion section.

Please see the discussion section.

  1. The possible generalization of the results should be debated.

Thank you for this comment. We included this point in the discussion section.

Please see the discussion section.

Reviewer 3 Report

Comments and Suggestions for Authors

Obturation Quality of Single-Cone Obturation Using Different Sizes of Matching Gutta-Percha Points of Two Reciprocating Single-File Systems in Curved and Straight Root Canals

The manuscript presents a well-structured and relevant investigation into the impact of different ISO sizes on obturation quality using two reciprocating single-file systems. The methodology is robust, employing a systematic approach to root canal preparation, obturation, and evaluation using fluorescence microscopy. The findings contribute valuable insights to endodontic practice, particularly in selecting appropriate ISO sizes for optimal obturation outcomes. However, the manuscript requires improvements in statistical justification, clinical interpretation of results, and methodological clarity.

1. Study Design and Methodology

Sample Selection and Justification:

The manuscript does not specify the age or source of the extracted teeth used. Since root canal anatomy can vary with age and other factors, clarifying whether teeth from younger or older individuals were included would enhance the generalizability of the study.

Standardization of Canal Morphology:

The study excludes canals with curvatures exceeding 20 degree, multiple canals, and irregularly shaped canals. While this ensures uniformity, it limits the applicability of findings to real-world clinical cases, where canal complexities are often encountered. A discussion on how these exclusions impact clinical relevance is needed.

Interobserver Reliability in Measurements:

The study does not mention how interobserver reliability was assessed when analyzing gutta-percha-filled areas (PGFA), sealer-filled areas (PSFA), and unfilled areas (PUA). Including intra- and inter-examiner agreement metrics (e.g., intraclass correlation coefficient) would strengthen reproducibility.

Irrigation Protocol Clarity:

The study alternates sodium hypochlorite and EDTA during preparation but does not specify whether the solutions were flushed between applications to prevent precipitate formation. This can affect sealer penetration and obturation quality. A clearer explanation of irrigation steps would be beneficial.

2. Statistical Analysis and Interpretation

Use of Kruskal-Wallis Test Without Justification:

The authors state that Kruskal-Wallis and Mann-Whitney U tests were used for statistical comparisons, but they do not provide normality test results (e.g., Kolmogorov-Smirnov test). If data were normally distributed, parametric tests like ANOVA would be more appropriate. Include justification for using nonparametric tests.

Confidence Intervals for Key Findings:

The study presents means and standard deviations but does not include confidence intervals. Reporting 95% confidence intervals would provide a clearer understanding of the variability in results.

Effect Size Consideration:

While statistical significance is reported, effect sizes (e.g., Cohen’s d) are not provided. Given the large sample size, small but statistically significant differences may not be clinically meaningful. Adding effect sizes would clarify the practical significance of the findings.

3. Results and Clinical Implications

Comparison with Other Obturation Techniques:

The study discusses findings in the context of single-cone obturation but does not compare them with other obturation methods like warm vertical compaction or thermoplastic techniques. Discussing how single-cone obturation compares to alternative techniques would enhance the clinical relevance.

Clinical Interpretation of PGFA, PSFA, and PUA Findings:

The study reports percentages of gutta-percha, sealer, and unfilled areas but does not explicitly discuss how these values translate to clinical success. For example, does a 1% increase in PUA significantly affect long-term prognosis? Providing clinical context would improve the discussion.

Apical Region Significance:

The study finds that larger ISO sizes result in better filling in the apical third, but it does not discuss the implications for apical microleakage or post-treatment outcomes. This should be addressed.

4. Figures and Tables

Table Formatting:

Some tables contain excessive decimal places (e.g., 87.0% instead of 87%). Standardizing decimal places would improve readability.

Figure Clarity:

Figures showing fluorescence microscopy images should be enhanced for contrast and labeled more clearly to indicate gutta-percha, sealer, and voids.

STROBE Compliance:

The study does not explicitly state adherence to STROBE guidelines for observational studies. Including a STROBE checklist would improve methodological transparency.

5. Language and Readability

Minor Grammatical Issues:

The manuscript has minor grammatical errors and awkward phrasing (e.g., Higher ISO sizes, particularly ISO 40 in Procodile® and ISO 45 in WaveOne® Gold, consistently demonstrated superior results). A thorough proofreading would improve fluency.

Redundant Phrasing:

Some sections repeat the same findings in different ways. Streamlining the discussion and results would enhance readability.

Clarification of Terminology:

The manuscript refers to “Procodile” and “WaveOne® Gold multiple times without clearly explaining their differences in design and function. A brief explanation would help readers unfamiliar with these systems.

Minor Comments (Optional Enhancements)

Add More Recent References:

While the manuscript cites relevant literature, incorporating more recent studies (2023 - 2024) on single-cone obturation techniques would strengthen the discussion.

Discuss Potential Impact of Sealer Type:

The study uses AH Plus, but alternative sealers (e.g., bioceramic sealers) may perform differently. A brief discussion of how sealer selection could influence the results would be beneficial.

Expand on Clinical Recommendations:

The study finds that larger ISO sizes improve obturation quality. Should clinicians always prepare canals to ISO 40 or 45, or does this depend on anatomical considerations? A short section on clinical recommendations would be useful.

Final Recommendation
Major Revisions

Comments on the Quality of English Language

Minor revisions required 

Author Response

Obturation Quality of Single-Cone Obturation Using Different Sizes of Matching Gutta-Percha Points of Two Reciprocating Single-File Systems in Curved and Straight Root Canals

Point by point response to all comments raised by Reviewer 3

Dear reviewer, thank you very much for your review and all valuable comments and recommendations to increase the quality of the submitted paper.

The manuscript presents a well-structured and relevant investigation into the impact of different ISO sizes on obturation quality using two reciprocating single-file systems. The methodology is robust, employing a systematic approach to root canal preparation, obturation, and evaluation using fluorescence microscopy. The findings contribute valuable insights to endodontic practice, particularly in selecting appropriate ISO sizes for optimal obturation outcomes. However, the manuscript requires improvements in statistical justification, clinical interpretation of results, and methodological clarity.

Thank you very much for this feedback and the review. We will address all points in the revised version of our manuscript.

  1. Study Design and Methodology

Sample Selection and Justification:

The manuscript does not specify the age or source of the extracted teeth used. Since root canal anatomy can vary with age and other factors, clarifying whether teeth from younger or older individuals were included would enhance the generalizability of the study.

Thanks a lot for your remark. We added the suggested information (age). The root canals were selected according their initial apical size. The section within the M&M section was clarified and the detailed information was added.

Please see. 2.1., 2.2 and 2.3 of the revised version. Lines 90-122.

  1. Standardization of Canal Morphology:

The study excludes canals with curvatures exceeding 20 degree, multiple canals, and irregularly shaped canals. While this ensures uniformity, it limits the applicability of findings to real-world clinical cases, where canal complexities are often encountered. A discussion on how these exclusions impact clinical relevance is needed.

We added the suggested discussion of this aspect in the discussion section.

Please see the rewritten discussion section.

  1. Interobserver Reliability in Measurements:

The study does not mention how interobserver reliability was assessed when analyzing gutta-percha-filled areas (PGFA), sealer-filled areas (PSFA), and unfilled areas (PUA). Including intra- and inter-examiner agreement metrics (e.g., intraclass correlation coefficient) would strengthen reproducibility.

The comment is correct and important for every not automatically investigation or intervention. We considered this aspect in a lot of own publications (e.g., Schwesig et al. 2010, 2014a, 2014b, 2018). In the present study, only one operator performed the evaluation of the 420 section. As part of this study, it was not possible to test intra- or interrater reliability. We added this aspect in the limitations as follows:

Line 335-337:   Furthermore, the study did not judge the influence of different investigators (interobserver reliability) concerning the analysis of gutta-percha-filled areas (PGFA), sealer-filled areas (PSFA), and unfilled areas (PUA)……

  1. Irrigation Protocol Clarity:

The study alternates sodium hypochlorite and EDTA during preparation but does not specify whether the solutions were flushed between applications to prevent precipitate formation. This can affect sealer penetration and obturation quality. A clearer explanation of irrigation steps would be beneficial.

Based on your valuable remark, we improved the methods and added this point additionally in the discussion sections follows:

Please see the corrected sections lines 150-162 and the rewritten discussion section.

  1. Statistical Analysis and Interpretation

Use of Kruskal-Wallis Test Without Justification:

The authors state that Kruskal-Wallis and Mann-Whitney U tests were used for statistical comparisons, but they do not provide normality test results (e.g., Kolmogorov-Smirnov test). If data were normally distributed, parametric tests like ANOVA would be more appropriate. Include justification for using nonparametric tests.

We added the results of the Kolmogorov-Smirnov test at the beginning of the results section as justification for using nonparametric tests:

Please lines 211-213: The Kolmogorov-Smirnov test showed that the PGFA, PSFA, and PUA did not have a normal distribution (P<0.001). Therefore, the data were analyzed using non-parametric tests.  Please see 2.6

  1. Confidence Intervals for Key Findings:

The study presents means and standard deviations but does not include confidence intervals. Reporting 95% confidence intervals would provide a clearer understanding of the variability in results.

Thank you very much for this helpful comments. In line with your suggestion, we have added 95% confidence intervals (95% CI) to tables 1, 2, 4, 5, and 6.

Please see tables 1, 2, 4, 5, 6 of the revised version.

  1. Effect Size Consideration:

While statistical significance is reported, effect sizes (e.g., Cohens d) are not provided. Given the large sample size, small but statistically significant differences may not be clinically meaningful. Adding effect sizes would clarify the practical significance of the findings.

At first, we are completely in line with you that clinical meaningfulness is much more important than statistical significance! The statistical significance is strongly dependent on the sample size, in contrast to the effect size (d or hp2). For this background, we added effect sizes to all relevant tables as suggested.

Please see tables and results section.

  1. Results and Clinical Implications

Comparison with Other Obturation Techniques:

The study discusses findings in the context of single-cone obturation but does not compare them with other obturation methods like warm vertical compaction or thermoplastic techniques. Discussing how single-cone obturation compares to alternative techniques would enhance the clinical relevance.

Thank you for this helpful comment. The main goal of the present investigation was to evaluate the different preparation protocols combined with matching gutta-percha points used for single-cone obturation technique. The comparison with other obturation techniques was added to the discussion section.

Please see the rewritten discussion section. This point was addressed.

  1. Clinical Interpretation of PGFA, PSFA, and PUA Findings:

The study reports percentages of gutta-percha, sealer, and unfilled areas but does not explicitly discuss how these values translate to clinical success. For example, does a 1% increase in PUA significantly affect long-term prognosis? Providing clinical context would improve the discussion.

Thanks a lot for the valuable hint. We improved the discussion in this direction.

Please see the rewritten discussion section. This point was addressed.

  1. Apical Region Significance:

The study finds that larger ISO sizes result in better filling in the apical third, but it does not discuss the implications for apical microleakage or post-treatment outcomes. This should be addressed.

We discussed this aspect as suggested:

Please see the rewritten discussion section.

  1. Figures and Tables

Table Formatting:

Some tables contain excessive decimal places (e.g., 87.0% instead of 87%). Standardizing decimal places would improve readability.

Thank you for this comment. All sections were checked and finally we used three reliable numbers in all tables and two reliable numbers in all figures in order to standardize the presentation and enhance the readability, especially with regard to the figures.

  1. Figure Clarity:

Figures showing fluorescence microscopy images should be enhanced for contrast and labeled more clearly to indicate gutta-percha, sealer, and voids.

Thank you for this comment. Figure 3a and b were improved as suggested.

Please see figure 3.

  1. STROBE Compliance:

The study does not explicitly state adherence to STROBE guidelines for observational studies. Including a STROBE checklist would improve methodological transparency.

Dear Reviewer, thank you for this comment. We thought about the STROBE guidelines while preparing the study protocol for the ethical approval and the institutional process of this work. We use the STROBE guidelines for clinical observational studies and cross-sectional as well as cohort studies. Therefore, we decided to organize the methodological details following the STROBE guidelines. We revised the paper regarding these points. We believe it was not necessary to include this point in a laboratory study.

  1. Language and Readability

Minor Grammatical Issues:

The manuscript has minor grammatical errors and awkward phrasing (e.g., Higher ISO sizes, particularly ISO 40 in Procodile® and ISO 45 in WaveOne® Gold, consistently demonstrated superior results). A thorough proofreading would improve fluency.

We improved the sentence as suggested:

Line 298-300:   Higher ISO values, particularly ISO 40 in Procodile® and ISO 45 in WaveOne® Gold, consistently showed better results (PGFA percentages: 87 and 85%).

Furthermore, we checked the whole manuscript regarding grammatical errors and awkward phrasing.

  1. Redundant Phrasing:

Some sections repeat the same findings in different ways. Streamlining the discussion and results would enhance readability.

Thank you for this important point. We checked and reworked the discussion regarding redundant phrasing.

  1. Clarification of Terminology:

The manuscript refers to “Procodile” and “WaveOne® Gold multiple times without clearly explaining their differences in design and function. A brief explanation would help readers unfamiliar with these systems.

We added a brief explanation in the methods section as suggested.

Please see the added explanation regarding the used file system in the M&M section. Lines 153-158.

  1. Minor Comments (Optional Enhancements)

Add More Recent References:

While the manuscript cites relevant literature, incorporating more recent studies (2023 - 2024) on single-cone obturation techniques would strengthen the discussion.

Dear reviewer, thank you for this important point. We added more recent studies to the revised version of the manuscript.

Please see the revised version of the manuscript- reference list.

  1. Discuss Potential Impact of Sealer Type:

The study uses AH Plus, but alternative sealers (e.g., bioceramic sealers) may perform differently. A brief discussion of how sealer selection could influence the results would be beneficial.

We included a short discussion regarding the influence of sealer selection for the results.

Please see the rewritten discussion section.

  1. Expand on Clinical Recommendations:

The study finds that larger ISO sizes improve obturation quality. Should clinicians always prepare canals to ISO 40 or 45, or does this depend on anatomical considerations? A short section on clinical recommendations would be useful.

This is an important point and we follow your considerations. As suggested, we restructured the entire discussion section and clinical recommendations. The conclusion was corrected and clarified as suggested.

Please see the discussion section and conclusion of the revised version.

Reviewer 4 Report

Comments and Suggestions for Authors

The manuscript addresses a modern and multidisciplinary research topic with an impact on the obturation quality of dental root canals from human permanent teeth. The study investigates the effect of selected ISO files sizes on obturation quality when using a single-file system in combination with its corresponding single-cone gutta-percha, employing the single-cone technique.  The originality of the research is represented by the fact it seeks to identify how preparation of root canals using different ISO file sizes affect filling quality in root canals with different morphologies such as curved or straight canals. The research conclusion demonstrates that ISO file size significantly influences obturation quality, with ISO 40 and ISO 45 producing the best results across both systems tested in various sections and configurations of root canals.

Author Response

Obturation Quality of Single-Cone Obturation Using Different Sizes of Matching Gutta-Percha Points of Two Reciprocating Single-File Systems in Curved and Straight Root Canals

Point by point response to all comments raised by Reviewer 4

Dear reviewer, thank you very much for your review and all valuable comments and recommendations to increase the quality of the submitted paper.

  1. The manuscript addresses a modern and multidisciplinary research topic with an impact on the obturation quality of dental root canals from human permanent teeth. The study investigates the effect of selected ISO files sizes on obturation quality when using a single-file system in combination with its corresponding single-cone gutta-percha, employing the single-cone technique. The originality of the research is represented by the fact it seeks to identify how preparation of root canals using different ISO file sizes affect filling quality in root canals with different morphologies such as curved or straight canals. The research conclusion demonstrates that ISO file size significantly influences obturation quality, with ISO 40 and ISO 45 producing the best results across both systems tested in various sections and configurations of root canals.

Strengths of manuscript

The manuscript corresponds to the stated purpose and objectives of the journal.

The title accurately reflects the content of the paper. It is relevant for the manuscript.

The abstract is structured as an accurate synopsis of the paper. It contains results which are presented and substantiated in the main text. 

The main conclusion mentioned in the abstract is not exaggerated.

The introduction presents in detail the current state of knowledge on the approached subject and highlights why this research is important.  It is define the purpose of the work and its significance. At the end of the Introduction chapter the proposed hypothesis and the null hypothesis of the research are presented. Also, the introduction includes 22 relevant references, out of which 8 are published after 2020. 

The manuscript contains a complex research method. The study obtained the ethical approval from “Ethics Committee of Martin Luther University Halle-Wittenberg, Halle, Germany (protocol number: 2024-023).” The all steps research and software used for statistical analysis are described in sufficient details to allow another researcher to reproduce the results. There are included 3 figures in chapter “Materials and Methods” in order to explain better sample selection, steps in root canal treatment and schematic representation of sections analysis under fluorescence microscope. Data were analyzed using descriptive statistics and statistical tests: Kolmogorov-Smirnov test, Kruskal-Wallis test, Mann-Whitney U test. The research method is described taking into account the research method used by another 4 studies.

The performed analyses are appropriate. The research results are presented at higher standards, including 6 tables and 3 figures. The results of the research are concisely and systematically presented. They are elaborated according to authors’ guideline.

The discussions present an interpretation of the results in perspective of previous studies and of the aim of the study. The findings and their implications are discussed. At the end of the discussion chapter, the limitations of the study and possible causes of affecting the validity of the findings are presented. The comparison of the research results was made taking into account 13 references, out of which 10 are published after 2020.

The conclusions are presented in a clearly manner and highlight the research implications for clinical practice and future research. The author mentioned that the findings emphasize the critical importance of ISO size selection in clinical practice to achieve high-quality root canal fillings, particularly in the apical region, where effective sealing is essential for long-term treatment success. They are interesting for the readership of the journal.

The references are in accordance with the studied topic. The manuscript contains 40 references, out of which 16 are published after 2020. 

Dear Reviewer, thank you very much for this feedback. We are really impressed and motivated to include all points raised during review round one. We added more references following the recommendation of all reviewers published after 2020. Please see the revised manuscript.

Weakness of the manuscript

It would be better if the authors will reformulate the title so that it doesn't contain the same word twice – “obturation”

We changed the title as suggested:

Line 2-4:          Obturation Quality of Single-Cone Obturation Using Different Sizes of Matching Gutta-Percha Points of Two Reciprocating Single-File Systems in Curved and Straight Root Canals

It would be better if the authors would shorten the abstract to contain 300 words, according to the instructions for authors. The current abstract contains 400 words.

We shortened the abstract as suggested. In the revised form the guidelines (a total of about 300 words) were respected. Thank you for this point.

It would be better if the authors will clarify if the study was performed on 140 roots canals or 140 roots, because there are roots with a single canal and roots with two canals.

         In chapter “2.3. Sample population” the authors mentioned “A total of 140 root canals were

         randomly divided into 14 groups”, but in the Figure 1, the authors mentioned “randomized 

         roots (n=140)”

Thank you for this helpful comment. We corrected the text and figure. The term “Root canals” is used throughout the entire manuscript.

Please see chapter 2.3 and figure 1.

It would be better if in the chapter ”2.5. Evaluation and statistical analysis”, the authors would mention the microscope objective used to analyze each section under the fluorescence microscope.

Thank you for this comment. The corrections were made in 2.5. and the missing information was added

The microscope objective (PlanApo 4.0 plus digital zoom 1.5) and resulting magnification of 6x was added. Please see chapter 2.5.

It would be better if the authors will include also H statistic and degree of freedom (df) in Kruskal-Wallis test results report and U statistic and z score in Mann-Whitney U test results report.

Thank you for this recommendation. In accordance to the other reviewers, the tables and results were corrected and missing information was added without exceeding the results to an unreadable extent.

Please see the tables and results section.

Round 2

Reviewer 1 Report

Comments and Suggestions for Authors

the authors have made the requested changes

Author Response

Dear Reviewer,

thank you very much for reviewing the study and manuscripts. Your comments helped to increase the sientific value of the paper.

Reviewer 2 Report

Comments and Suggestions for Authors

Dear authors, I have no further questions.

Author Response

Dear Reviewer

Thank you very much for reviewing the manuscripts and all the valuable comments increasing the scientific value of the paper.

Reviewer 3 Report

Comments and Suggestions for Authors

Congratulations to the authors and thank you for the resubmission. The revision has substantially increased the readability credential of the manuscript.

i recommend acceptance.

Author Response

Dear Reviewer,

Thank you very much for reviewing the manuscripts and your feedback. Your comments and recommendations were helpful and increased the scientific value of the paper.